# Emotion regulation success involves systematic gradient-based reconfigurations of large-scale activation patterns in the human brain

Ruien Wang[1]*, Remi Janet[1,2], Carmen Morawetz[3☯], Anita Tusche[1,4☯]*

**1** Department of Psychology, Queen's University, Kingston, Canada, **2** Institut des Sciences Cognitives Marc Jeannerod, UMR-5229 CNRS, Bron, France, **3** Department of Psychology, University of Innsbruck, Innsbruck, Austria, **4** Center for Neuroscience Studies, Queen's University, Kingston, Canada

☯ These authors contributed equally to this work (shared last authorship).
* ruien.wang@queensu.ca (RW); anita.tusche@queensu.ca (AT)

## Abstract

Emotion regulation is essential for well-being and mental health, yet individuals vary widely in their emotion regulation success. Why? Traditional neuroimaging studies of emotion regulation often focus on localized neural activity or isolated networks, overlooking how large-scale brain organization relates to the integration of distributed systems and sub-processes supporting regulatory success. Here, we applied a novel system-level framework based on spatial gradients of macroscale brain organization to study variance in emotion regulation success. Using two large functional magnetic resonance imaging (fMRI) datasets ($n = 358$, $n = 263$), we projected global activation patterns from a laboratory emotion regulation task onto principal gradients derived from independent resting-state fMRI data from the Human Connectome Project. These gradients capture low-dimensional patterns of neural variation, providing a topographical framework within which complex mental phenomena, such as emotion regulation, emerge. In both datasets, individual differences in regulation success were associated with systematic reconfiguration along Gradient 1—a principal axis differentiating unimodal and heteromodal brain areas. This gradient-based neural reconfiguration also associates with lower negative affect in daily life, as measured via smartphone-based experience sampling in a subset of participants ($n = 55$). Meta-analytic decoding via Neurosynth revealed that Gradient 1 and regulation success align with multiple psychological processes, including social cognition, memory, attention, and negative emotion, suggesting this gradient reflects diverse, integrative demands during effective emotion regulation. These findings introduce a gradient-based perspective on emotion regulation success that is biologically grounded in well-established large-scale brain organization and ecologically valid through its links with real-world emotional experience. Such gradient-based dynamics may serve as predictive biomarkers of regulatory success and inform targeted interventions in clinical populations.

**Data availability statement:** Relevant data and analysis code supporting the key findings of this study have been made available on the Open Science Framework (OSF, https://osf.io/yk85c/).

**Funding:** This work was supported by National Science and Engineering Research Council of Canada (NSERC, to AT) and the Social Sciences and Humanities Research Council of Canada (SSHRC, to AT). These funders played no role in the study design, data collection and analysis, decision to publish, or preparation of the manuscript.

**Competing interests:** The authors have declared that no competing interests exist.

**Abbreviations:** AIC, akaike information criterion; DGP, German Psychological Society; DMN, default mode network; DS, discovery; sampleED, euclidean distance; ERQ, emotion regulation questionnaire; FDR, false discovery rate; FP, frontoparietal; fMRI, functional magnetic resonance imaging; GLM, group-level GLM; HCP, human connectome project; HRF, hemodynamic response function; HSREB, Health Sciences and Affiliated Teaching Hospitals Research Ethics Board; ISI, inter-stimulus interval; LMM, linear mixed models; MNI, Montreal Neurological Institute; NiMARE, Neuroimaging Meta-Analysis Research Environment; RS, replication sample; SPM, statistical parametric mapping.

## Introduction

The ability to regulate the intensity and duration of emotional experiences is a cornerstone of adaptive behavior in dynamic environments, influencing decision-making [1–4], and both physical and mental well-being [5–8]. Yet individuals differ widely in their emotion regulation success—variability that shapes vulnerability to stress and psychopathology [9]. Understanding the neural basis of emotion regulation and the sources of individual differences in regulatory success has thus become a central goal in affective neuroscience [10].

Traditional functional magnetic resonance imaging (fMRI) studies and meta-analyses have provided important insights into the neural mechanisms supporting emotion regulation across diverse populations and task paradigms [11–15]. These studies consistently demonstrate that emotion regulation engages coordinated activity across large-scale cortical and subcortical networks [16–23], including frontal, parietal, and temporal regions—such as the dorsolateral and ventrolateral prefrontal cortices, supplementary motor area, and inferior parietal lobule—as well as subcortical structures like the amygdala, ventral striatum, and periaqueductal gray. These areas are implicated in a broad range of cognitive and affective processes, including emotional reactivity, attention, working memory, cognitive control, and language [18,20,24,25]. Collectively, these findings challenge simplistic, region-specific models, instead positioning emotion regulation as a complex phenomenon that relies on a spatially distributed suite of diverse brain areas embedded in multiple large-scale networks.

Building on this perspective, research on functional and effective connectivity has revealed dynamic, context-dependent interactions among occipital, subcortical, and frontal regions during emotion regulation [26–29]. These findings challenge classical top-down models [1,15,23,30–32] and highlight the flexible nature of distributed neural configurations underlying regulatory processes. However, while such studies have begun to map the connectivity between individual areas or networks underlying emotion regulation, relatively few have been able to directly link these patterns to individual differences in successful regulation outcomes [27]. Moreover, traditional analytic approaches still often emphasize isolated peak activations or predefined networks underlying variance in emotion regulation success. Such methods frequently overlook the brain's topographical and hierarchical organization [30,31]—principles essential for understanding how complex cognitive functions like emotion regulation might emerge from the dynamic integration of distributed neural systems.

To address this gap and adopt a more holistic approach to emotion regulation across the *entire* brain, we use a gradient-based systems neuroscience framework. Gradients (or manifolds) provide low-dimensional representations of functional or structural properties along continuous axes, capturing fundamental principles of macroscale topographical architecture and revealing how regions and networks are embedded within broader hierarchical processing streams. Gradients have been reliably identified across species and brain modalities [33–40] and are thought to provide an intrinsic coordinate system that constrains the brain's dynamic repertoire [41]. Principal gradients track a hierarchy from primary sensory processing to high-order functions such as emotion, language, and social cognition, offering a

structural scaffold for hierarchical information processing [35,42]. Multimodal neuroimaging work shows that gradients capture functional diversity across cognitive tasks, underscoring the role of large-scale brain organization in shaping brain function [43]. Evidence from epilepsy demonstrates that gradients sensitively track reorganization of hippocampal and neocortical systems during episodic memory states [44]. Alterations in large-scale brain organization and hierarchy are also observed in autism, accounting for diverse symptomology and neurodevelopmental features [45–47]. The gradient framework further aligns with theoretical accounts of allostasis, which propose that the brain predicts and regulates internal bodily states through coordinated, whole-brain organization [48,49]. Within this view, hierarchical gradients may support domain-general neural functions, offering insight into how emotional processing and other cognitive phenomena may reflect whole-brain, allostatic computations rather than distinct, localized processes. Recent work has shown that projecting task-evoked neural activity into gradient space can capture individual differences in self-regulation, such as during dietary decision-making [50]. Although not yet applied to emotion regulation, this gradient framework holds promise for understanding how regulatory processes unfold within the constraints of the brain's intrinsic topographical landscape.

Building on this foundation, we investigated whether individual differences in emotion regulation success can be explained by brain-wide shifts in activation patterns aligned with the brain's principal functional gradients. In other words, we hypothesized that neural responses associated with effective emotion regulation involve *systematic* reconfigurations along principal axes of the brain's macroscale architecture. To test this notion, we analyzed two large fMRI datasets ($n=358$, $n=263$) using a standard emotion regulation task in which participants used cognitive reappraisal to regulate their emotional experiences. As the most widely studied and validated emotion regulation strategy [11], cognitive reappraisal provides an effective context for evaluating our gradient framework approach to emotion regulation. Participants either passively viewed negative stimuli ("Look") or regulated their emotions using reappraisal ("Regulate") (Fig 1A). The analyses followed four steps. First, we mapped global activation patterns from both conditions onto principal gradients derived from task-free resting-state fMRI data in a large, independent Human Connectome Project (HCP) sample [35,51] (Fig 2A and 2B). Thus, these principal gradients of macroscale topographical organization were completely independent of externally elicited emotional experiences, reappraisal, or emotion regulation more generally. In our study, we focused on mapping task-evoked global activation patterns onto the first five principal gradients that explained most of the variance (~55%) in the independent HCP data. These five dimensions of brain variation differentiate unimodal and heteromodal (association) cortices (Gradient 1), visual and sensorimotor cortices (Gradient 2), the default mode network (DMN) and frontoparietal systems (FP) (Gradient 3), ventral and dorsal attention networks (Gradient 4), and visual cortex and ventral attention networks (Gradient 5) (Fig 2B).

Second, we assessed whether individual differences in emotion regulation success in our two fMRI datasets correspond to shifts of projected task-evoked global activation patterns along these principal gradients (Fig 2C). Based on prior evidence that reappraisal engages heteromodal (association cortex) regions [18,52], we predicted a regulatory role of systematic shifts along Gradient 1, which separates unimodal from heteromodal cortices.

Third, to assess the ecological validity of our findings, we next examined whether regulation-related neural gradient shifts are also associated with affective outcomes in daily life. Using smartphone-based experience sampling from a subset of participants ($n=55$), we tested whether gradient-based reconfigurations of activation patterns that correspond to laboratory regulation success are also associated with more positive affect in real-world settings, consistent with improved affective outcomes for better regulators.

Finally, we used meta-analytic decoding with Neurosynth [53] to link brain activation patterns and gradients related to variance in emotion regulation success to established cognitive constructs, providing a theory-neutral interpretation of the underlying psychological processes.

By projecting global activation patterns evoked by emotion generation and regulation into gradient space, our study offers a novel perspective on the spatially distributed neural mechanisms of emotion regulation. Building on evidence that reappraisal engages heteromodal regions and on the principle that Gradient 1 captures a hierarchy from unimodal

PLOS Biology

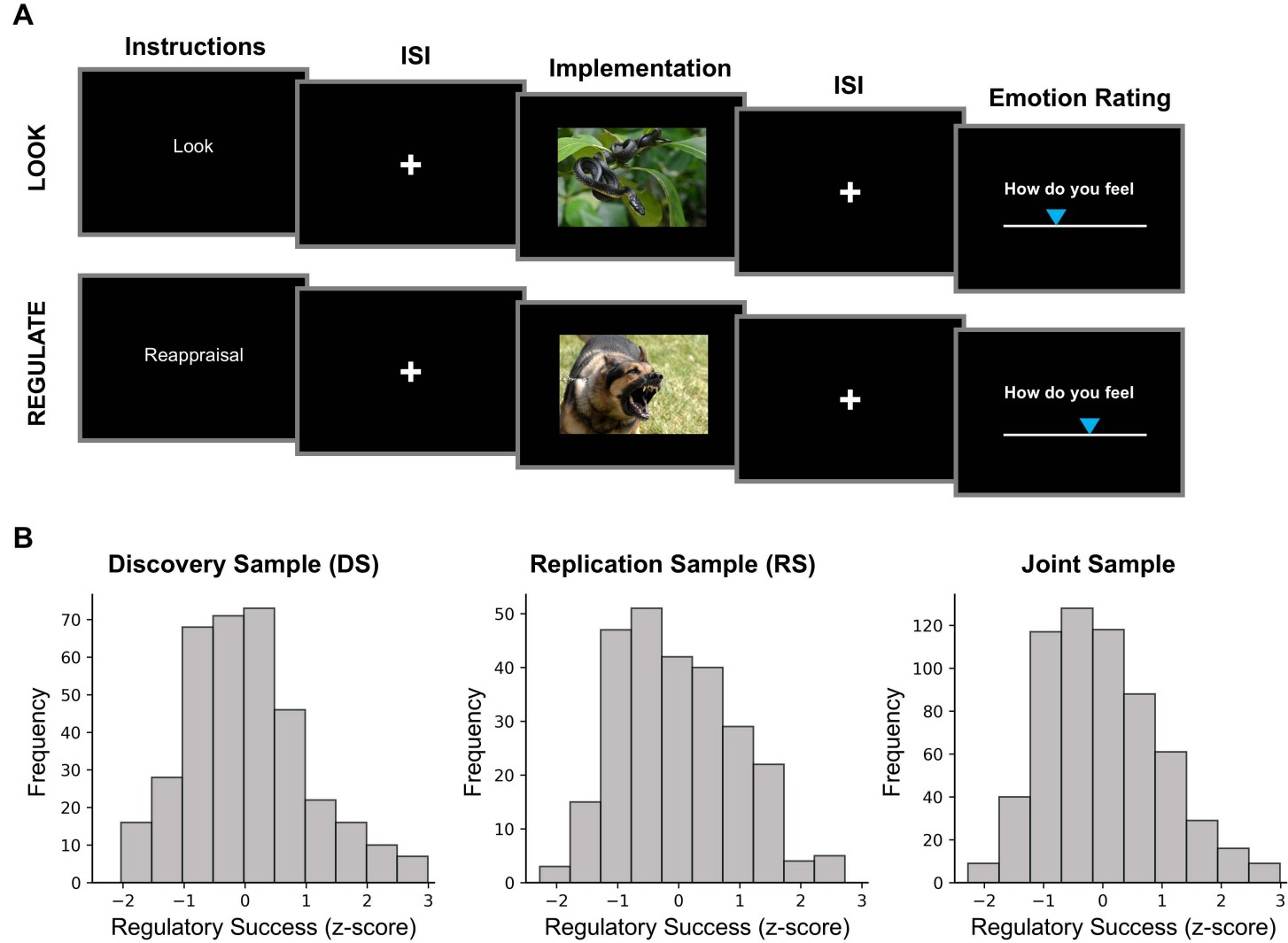

**Fig 1. Emotion regulation task and distribution of regulatory success scores. A.** On each trial, participants were cued to either passively look at negative images ("Look") or actively down-regulate negative emotions using cognitive reappraisal ("Regulate"). After a jittered fixation period (inter-stimulus interval, ISI), a negative image was presented (implementation phase) and participants subsequently rated their emotional state. **B.** Distributions of emotion regulation success scores in the discovery (DS), replication (RS), and joint samples. For each participant, regulatory success was calculated as the mean difference between emotional state ratings in all trials of the regulate vs. look conditions, with higher positive scores indicating greater regulatory success. Scores were then z-normalized within each study of both samples to account for differences in rating scales across sub-samples (S1 Table). Figure includes open-source images retrieved from Wikimedia Commons: Military dog (https://commons.wikimedia.org/wiki/File:Military_dog_barking.JPG) and bird-eating snake (https://commons.wikimedia.org/wiki/File:Bird-eating_snake_(7607449358).jpg). The data and code used to generate the figure can be found at: https://osf.io/yk85c/.

to heteromodal cortices, we hypothesized that emotion regulation would involve shifts along this principal gradient. We further predicted that these gradient shifts would track variance in regulatory success in the laboratory and reduced negative affect in daily life. This approach moves beyond region- and network-specific models, revealing that individual differences in emotional control stem from the systematic reorganization of large-scale activation patterns along core topographical axes of macroscale brain organization. Our findings suggest that variability in emotion regulation success reflects not only which regions are engaged, but how brain-wide functional dynamics are reconfigured within the brain's hierarchical

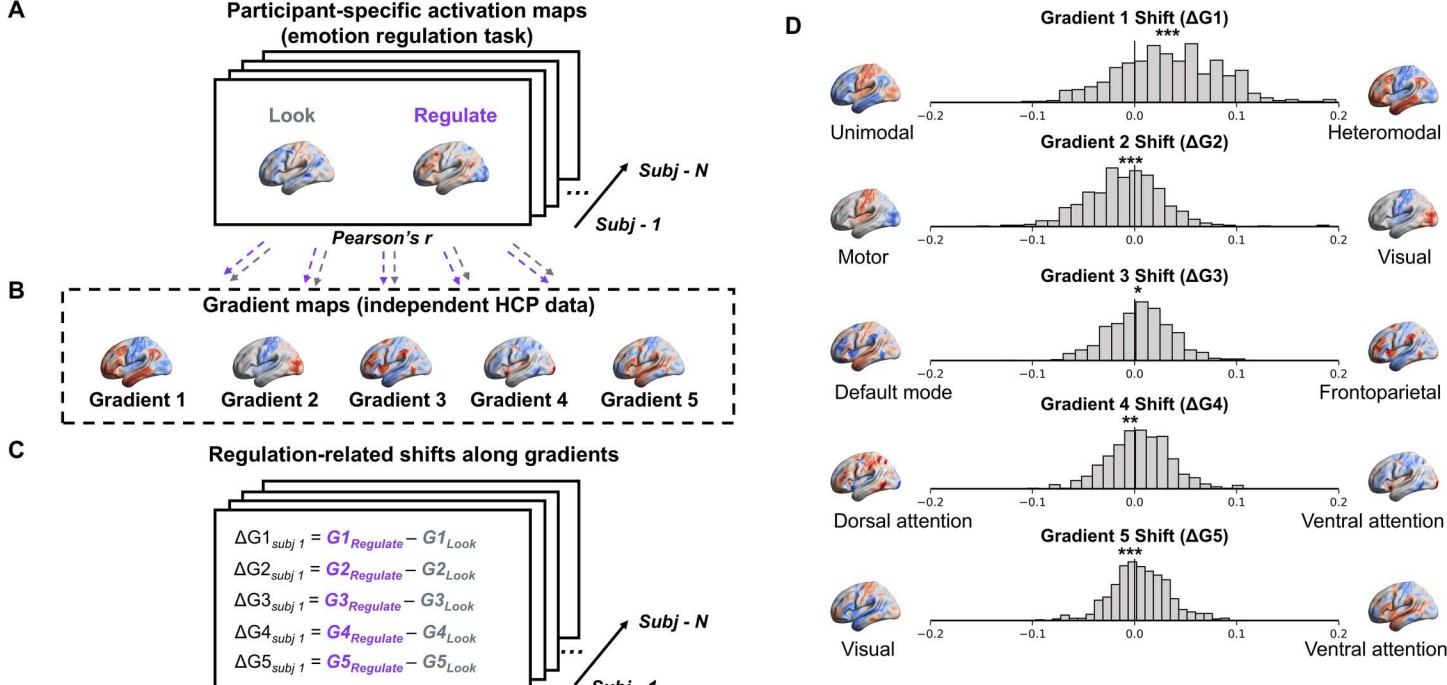

**Fig 2. Emotion regulation-related shifts in global activation patterns along principal gradients. A.** Participant's condition-specific activation maps ("Look" and "Regulate") were projected into gradient space by calculating the dot product with each canonical gradient map (Fisher z-transformed Pearson's r), yielding two similarity scores for each gradient per participant (one per task condition). **B.** Principal gradients were derived from fMRI resting-state data from an independent cohort (Human Connectome Project, HCP) [35], unrelated to emotion regulation or task-evoked emotional experiences. **C.** Emotion regulation-related shifts along each gradient were calculated as within-participant differences in similarity scores between projected global activation patterns for "Regulate" and "Look" conditions for each gradient (ΔG1 to ΔG5), with higher values indicating greater neural reconfiguration along a particular principal gradient. These change scores were then used as predictors of variance in task-based emotion regulation success. **D.** Distributions of emotion regulation-related shifts across gradients (ΔG1 to ΔG5). Asterisks indicate significant differences from zero along a particular gradient (Wilcoxon signed-rank tests, *$p < 0.05$; **$p < 0.01$; ***$p < 0.001$; Bonferroni-corrected). Brain images were plotted using BrainNet Viewer [58]. The data and code used to generate the figure can be found at: https://osf.io/yk85c/.

topographical architecture. These insights have broad implications for understanding neuropsychiatric vulnerability, well-being, and decision-making—domains in which emotion regulation plays a pivotal role.

## Results

### Behavioral emotion regulation success in a laboratory emotion regulation task

To assess individual differences in emotion regulation success, we analyzed behavioral data from an established emotion regulation task in two large samples: a discovery sample (DS, $n = 358$; 176 female, age: $42.04 \pm 6.96$; Mean ± SD) [54] and a replication sample (RS, $n = 263$, 156 female, age: $25.22 \pm 7.78$) [2,3, 55–57]. In each trial, participants either passively viewed negative images ("Look") or used cognitive reappraisal to down-regulate negative emotional experiences ("Regulate") (Fig 1A). After each trial, participants reported their emotional state on a rating scale. Ratings provided a behavioral index of regulatory success—defined as the difference between ratings in the regulate and look conditions, with higher positive values indicating more effective regulation of task-evoked emotional states. To account for variations in rating scales across individual studies, regulatory success scores were z-normalized per study in each sample. Fig 1B displays the distributions of normalized regulatory success scores in each sample (DS: $0.01 \pm 0.98$; RS: $-0.02 \pm 0.97$; Mean ± SD) and the joint sample ($-0.002 \pm 0.98$), with raw score distributions visualized in S1 Fig and reported in S1 Table. Regulatory

 

success varied widely across individuals in both samples, highlighting substantial inter-individual differences in emotion regulation success and motivating further investigation into their underlying neural mechanisms.

### Characterizing emotion regulation as systematic changes in whole-brain activation patterns along principal gradients of macroscale brain organization

We next investigated whether emotion regulation systematically reconfigures whole-brain activation patterns along principal gradients of large-scale brain organization. Specifically, we examined task-evoked activation patterns within the first five principal gradients derived from an independent HCP dataset, which together explain ~55% of variance and differentiate major functional systems—unimodal and heteromodal association cortices (Gradient 1), visual and sensorimotor cortices (Gradient 2), default mode network and frontoparietal systems (Gradient 3), ventral and dorsal attention networks (Gradient 4), and visual cortex and ventral attention network (Gradient 5) (Fig 2B). To this end, following prior work [50], we projected each participant's whole-brain activation maps—derived from condition-specific contrasts in the emotion regulation task (Fig 1A) against the implicit baseline—into gradient space. Specifically, we calculated Pearson correlation coefficients (Fisher Z-transformed) between each participant's condition-specific global activation map (unthresholded) during all trials of the look or regulate condition, respectively, and independently derived, established gradient maps obtained from fMRI resting-state data in the HCP [35] (Fig 2B). This approach produced two similarity scores per participant per gradient, reflecting how closely each condition-specific global activation map from the emotion regulation task ("Regulate" or "Look") aligned with five principal organizational gradients. The descriptive statistics of these condition-specific projections onto gradients are summarized in Table 1 and visualized in S2 Fig. For brevity, for this initial step that sets the stage for our key analyses (i.e., regulation-related shifts along these gradients), we focus on the joint sample for visualization ($n = 619$, see Table 1 for sample-specific information).

Condition-wise statistical tests confirmed that projected activation patterns for both the "Look" and "Regulate" conditions significantly differed from zero along all gradients (Wilcoxon signed-rank tests, all $p's < 0.001$, Bonferroni-corrected) [59], except Gradient 4 of the joint sample. Across both samples, global brain states evoked during either condition of the emotion regulation task were consistently characterized as more unimodal than heteromodal (Gradient 1), more visual than motor (Gradient 2), more aligned with the default mode network than frontoparietal control regions (Gradient 3), and

**Table 1. Descriptive statistics of projected task-evoked activation patterns along principal gradients.**

| Condition | Gradient 1 | Gradient 2 | Gradient 3 | Gradient 4 | Gradient 5 |
|---|---|---|---|---|---|
| *Joint sample* | | | | | |
| Look | −0.17±0.13 [−0.46, 0.25] | 0.41±0.23 [−0.11, 0.89] | −0.02±0.08 [−0.21, 0.19] | −0.0003±0.08 [−0.22, 0.24] | −0.07±0.06 [−0.22, 0.26] |
| Regulate | −0.13±0.14 [−0.47, 0.21] | 0.40±0.23 [−0.14, 0.88] | −0.02±0.08 [−0.22, 0.22] | 0.004±0.08 [−0.20, 0.21] | −0.06±0.06 [−0.22, 0.15] |
| *Discovery sample (DS)* | | | | | |
| Look | −0.10±0.11 [−0.37, 0.25] | 0.24±0.11 [−0.11, 0.52] | −0.02±0.07 [−0.21, 0.19] | −0.04±0.06 [−0.22, 0.14] | −0.06±0.07 [−0.22, 0.26] |
| Regulate | −0.06±0.11 [−0.38, 0.21] | 0.23±0.11 [−0.14, 0.47] | −0.03±0.07 [−0.21, 0.22] | −0.03±0.06 [−0.20, 0.18] | −0.06±0.07 [−0.22, 0.15] |
| *Replication sample (RS)* | | | | | |
| Look | −0.25±0.10 [−0.46, 0.06] | 0.64±0.13 [0.34, 0.89] | −0.08±0.05 [−0.21, 0.09] | 0.05±0.06 [−0.13, 0.24] | −0.07±0.06 [−0.19, 0.13] |
| Regulate | −0.23±0.12 [−0.47, 0.15] | 0.63±0.13 [0.32, 0.88] | −0.07±0.05 [−0.22, 0.10] | 0.05±0.06 [−0.20, 0.21] | −0.07±0.06 [−0.21, 0.15] |

*Note.* Values represent means and standard deviations, with observed ranges [min, max] in brackets.

more visual than ventral attention (Gradient 5). This was true for the joint sample (S2 Fig and Table 1) and each independent sample (Table 1). For Gradient 4, results point to sample-specific differences, with task-evoked activation patterns being more similar to the dorsal attention than ventral attention in DS, but not for participants in the RS group (Table 1).

Gradient-wise statistical tests further revealed significant shifts in the projected similarity of regulation-related global activation maps relative to the unregulated "Look" condition along principal gradients of hierarchical brain organization (Wilcoxon signed-rank tests of projected "Regulate" versus "Look" distributions per gradient). Specifically, for the joint sample, regulation of emotional experiences was associated with a significant shift toward more heteromodal brain states along Gradient 1 ($p < 0.001$), consistent with prior evidence highlighting the engagement of heteromodal cortices during emotion regulation [18,52] (Fig 2D). Additionally, regulation-related activation patterns became more motor-oriented (Gradient 2, $p < 0.001$), more frontoparietal (Gradient 3, $p = 0.002$), and more aligned with ventral attention networks on Gradients 4 and 5 (Gradient 4, $p = 0.006$; Gradient 5, $p < 0.001$; all $p$'s are Bonferroni-corrected for the multiple tests across 5 gradients × 3 samples [DS, RS, and joint]). Full statistical results for both separate samples are provided in S3 Table.

## Systematic shifts of global activation patterns along principal gradients reflect individuals' regulatory success

Having shown that emotion regulation (reappraisal) reconfigures whole-brain activation patterns along principal gradients of topographical brain organization, we next turn to our key question: are gradient-based shifts in large-scale activation patterns associated with individual differences in emotion regulation success? To probe the robustness of this core test of our study, we focus on sample-specific results in this section. Specifically, we applied separate linear mixed models (LMM) for each sample (DS, RS), using participant-specific regulatory success scores as the dependent variable. Predictors included individuals' regulation-related changes in gradient similarity relative to the "Look" condition (Regulate - Look) for each of the five principal gradients (ΔG1 to ΔG5), along with covariates for participants' average emotional reactivity (average emotional rating during all "Look" trials; see S1 Table for sample-specific descriptive statistics), age, sex, and study ID within each sample (see Eq. 1, Materials and methods).

In both independent samples, the model significantly explained variance in regulatory success (DS: $R^2 = 0.25$, $p = 0.01$; RS: $R^2 = 0.25$, $p = 0.002$). Notably, larger shifts along Gradient 1—which captures the transition from unimodal to heteromodal brain regions—were consistently associated with greater regulatory success (DS: $\beta = 3.15$, SE $= 0.88$, $t = 3.57$, 95% CI [1.47, 4.90], $p < 0.001$; RS: $\beta = 4.12$, SE $= 1.09$, $t = 3.77$, 95% CI [2.01, 6.23], $p < 0.001$) (Fig 3A). In contrast, shifts along Gradients 2–5 did not significantly covary with individuals' success in downregulating negative emotions (all $p$'s $> 0.19$; see Table 2 for details). There was also no significant effect of the control variables (i.e., age, sex, or study ID); but participants' average emotional reactivity (in "Look" trials) was associated with variance in regulatory success (all $p$'s $< 0.001$, Table 2). For completeness, Table 2 also provides details regarding the supplemental LMM results in the joint sample. This finding links systematic reconfigurations along a principal axis of topographical brain organization—Gradient 1—to variance in people's emotion regulation success, underscoring how the regulatory success of emotional experiences can emerge from constraints of spatial organization in the human brain.

As a sanity check, we also examined how neural activation during emotion regulation (Regulate versus Look) covaries with shifts along Gradient 1, using a traditional univariate analysis approach. To this end, we conducted a standard second-level analysis (General Linear Model, GLM) across the joint sample ($n = 619$) using SPM12 (https://www.fil.ion.ucl.ac.uk/spm/software/spm12/). For each participant, we entered the contrast image for Regulate > Look into a group-level GLM and included the participant-specific Gradient 1 shift (ΔG1) as a covariate. We then estimated positive and negative covariate contrasts, thresholded at $p < 0.001$ (uncorrected) at the voxel level and FWE-corrected at the cluster-level at $p < 0.05$. As expected, larger Gradient 1 shifts during emotion regulation were associated with greater activation in heteromodal regions and reduced activation in unimodal regions (see S4 Table for full details, S3 Fig for a visualization, and the OSF repository for the unthresholded $t$-map [https://osf.io/yk85c/]).

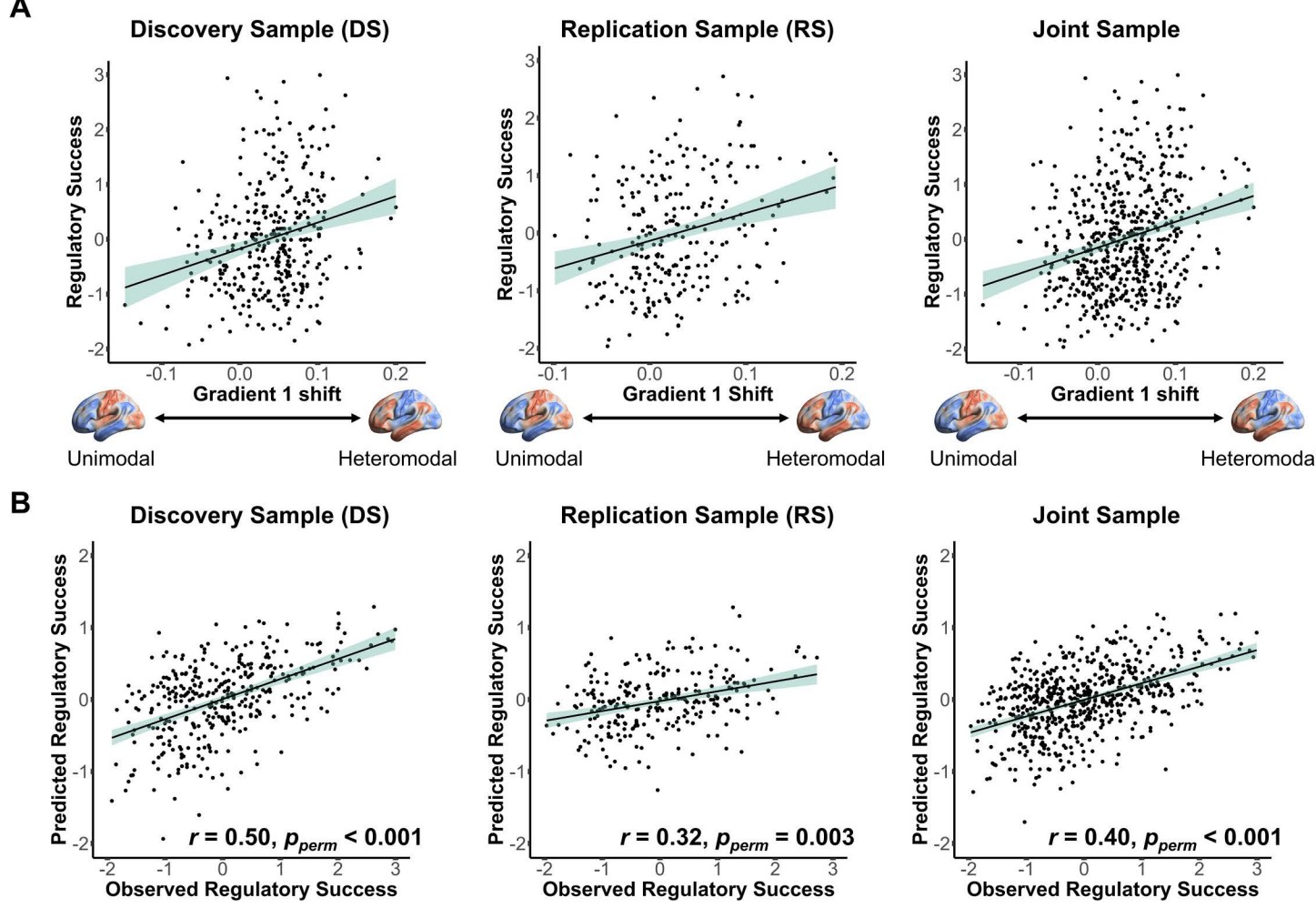

**Fig 3. Individual differences in emotion regulation success are associated with reconfigurations of global activation patterns along Gradient 1. A.** Greater shifts in whole-brain activation maps along Gradient 1 (from unimodal to heteromodal) during "Regulate" relative to "Look" conditions correspond to higher emotion regulation success in both independent samples (DS: $\beta = 3.15$, SE = 0.88, $t = 3.57$, 95% CI [1.47, 4.90], $p < 0.001$; RS: $\beta = 4.12$, SE = 1.09, $t = 3.77$, 95% CI [2.01, 6.23], $p < 0.001$). Results for the joint sample are also shown for comparison ($\beta = 3.77$, SE = 0.69, $t = 5.50$, 95% CI [2.45, 5.11], $p < 0.001$). **B.** Pearson correlations between observed and predicted regulatory success scores (z-scored) using leave-one-participant-out cross-validation in DS ($r = 0.50$, $p_{perm} < 0.001$, 95% CI [0.42, 0.58]), RS ($r = 0.32$, $p_{perm} = 0.003$, 95% CI [0.21, 0.43]), and the joint sample ($r = 0.40$, $p_{perm} < 0.001$, 95% CI [0.34, 0.47]). The significance of out-of-sample predictions was assessed via permutation tests (Bonferroni-corrected across tests in DS, RS, and the joint sample). Brain images were plotted using BrainNet Viewer [58]. The data and code used to generate the figure can be found at: https://osf.io/yk85c/.

### Gradient-based out-of-sample prediction of variance in regulatory success

Next, we evaluated the predictive power of our main model (Eq. 1) using leave-one-participant-out cross-validation within the DS, RS, and (for completeness) the joint samples. Predictive accuracy—assessed via correlations between observed and predicted emotion regulation success—was significant (DS: $r = 0.50$, 95% CI [0.42, 0.58], $p_{perm} < 0.001$; RS: $r = 0.32$, 95% CI [0.21, 0.43], $p_{perm} = 0.003$; joint sample: $r = 0.40$, 95% CI [0.34, 0.47], $p_{perm} < 0.001$, all p-values based on nonparametric permutation tests, Bonferroni-corrected [59]) (Fig 3B; see S4 Fig for study-specific results further highlighting the robustness across sub-samples).

**Table 2. Systematic reconfigurations of global activation maps along principal gradient 1 reflect individuals' regulatory success in a standard emotion regulation task.**

| Predictor | Estimate (SE) | 95% CI | t | p |
|---|---|---|---|---|
| *Joint sample* | | | | |
| Gradient 1 shift (ΔG1) | 4.38 (0.71) | [3.00, 5.75] | 6.16 | <0.001 |
| Gradient 2 shift (ΔG2) | 1.06 (0.95) | [−0.89, 2.84] | 1.11 | 0.26 |
| Gradient 3 shift (ΔG3) | −1.06 (1.11) | [−3.19, 1.12] | −0.96 | 0.34 |
| Gradient 4 shift (ΔG4) | −1.53 (1.17) | [−3.77, 0.76] | −1.31 | 0.19 |
| Gradient 5 shift (ΔG5) | 1.55 (1.31) | [−1.02, 4.10] | 1.18 | 0.24 |
| Emotional reactivity | 0.28 (0.04) | [0.21, 0.35] | 7.58 | <0.001 |
| Age | −0.003 (0.005) | [−0.01, 0.01] | −0.68 | 0.50 |
| Sex | −0.04 (0.08) | [−0.19, 0.11] | −0.53 | 0.60 |
| *Discovery sample (DS)* | | | | |
| Gradient 1 shift (ΔG1) | 3.15 (0.88) | [1.47, 4.90] | 3.57 | <0.001 |
| Gradient 2 shift (ΔG2) | 0.68 (1.27) | [−1.85, 3.08] | 0.54 | 0.59 |
| Gradient 3 shift (ΔG3) | −0.38 (1.35) | [−3.00, 2.23] | −0.28 | 0.78 |
| Gradient 4 shift (ΔG4) | −0.93 (1.48) | [−3.79, 1.93] | −0.63 | 0.53 |
| Gradient 5 shift (ΔG5) | 0.86 (1.59) | [−2.20, 3.98] | 0.54 | 0.59 |
| Emotional reactivity | 0.47 (0.05) | [0.38, 0.56] | 10.29 | <0.001 |
| Age | −0.01 (0.01) | [−0.02, 0.01] | −0.79 | 0.43 |
| Sex | 0.03(0.09) | [−0.15, 0.20] | 0.30 | 0.76 |
| *Replication sample (RS)* | | | | |
| Gradient 1 shift (ΔG1) | 4.12 (1.09) | [2.01, 6.23] | 3.77 | <0.001 |
| Gradient 2 shift (ΔG2) | 1.04 (1.41) | [−1.67, 3.75] | 0.74 | 0.46 |
| Gradient 3 shift (ΔG3) | −2.34 (1.81) | [−5.83, 1.13] | −1.30 | 0.19 |
| Gradient 4 shift (ΔG4) | −0.20 (1.78) | [−3.64, 3.24] | −0.11 | 0.91 |
| Gradient 5 shift (ΔG5) | 2.74 (2.14) | [−1.38, 6.86] | 1.28 | 0.20 |
| Emotional reactivity | 0.17 (0.06) | [0.15, 0.38] | 4.59 | < 0.001 |
| Age | 0.002 (0.01) | [−0.01, 0.02] | 0.35 | 0.73 |
| Sex | −0.06 (0.13) | [−0.31, 0.19] | −0.47 | 0.64 |

Moreover, to demonstrate cross-sample generalizability, we trained the model on DS data and tested its predictive performance on the independent RS data. The model significantly predicted variance in regulatory success in the RS group ($r = 0.37$, 95% CI [0.26, 0.47], $p_{perm} < 0.001$, permutation test, Figure S5). This out-of-sample performance—despite differences in task design, imaging parameters, and participant characteristics—highlights the robustness and generalizability of Gradient 1 shifts as a neural marker of individual differences in emotion regulation success.

### Similarity of decoded mental processes for gradient 1 and regulatory success-related global activation maps

Next, we identified the neural activation map associated with individual differences in regulatory success in the emotion regulation task, using a traditional univariate analysis approach. Specifically, we estimated a group-level GLM using participants' task-based regulatory success scores as a covariate for the "Regulate> Look" contrast in the joint sample ($n = 619$). The resulting unthresholded $t$-map for the covariate of regulatory success is shown in Fig 4A (also available on OSF https://osf.io/yk85c/; S5 Table provides a summary of thresholded results at $p < 0.001$, uncorrected).

To interpret the psychological relevance of this global activation map and its relationship to independently derived Gradient 1 [35] (Fig 4B), we applied meta-analytic functional decoding using Neurosynth [53]. This technique correlates

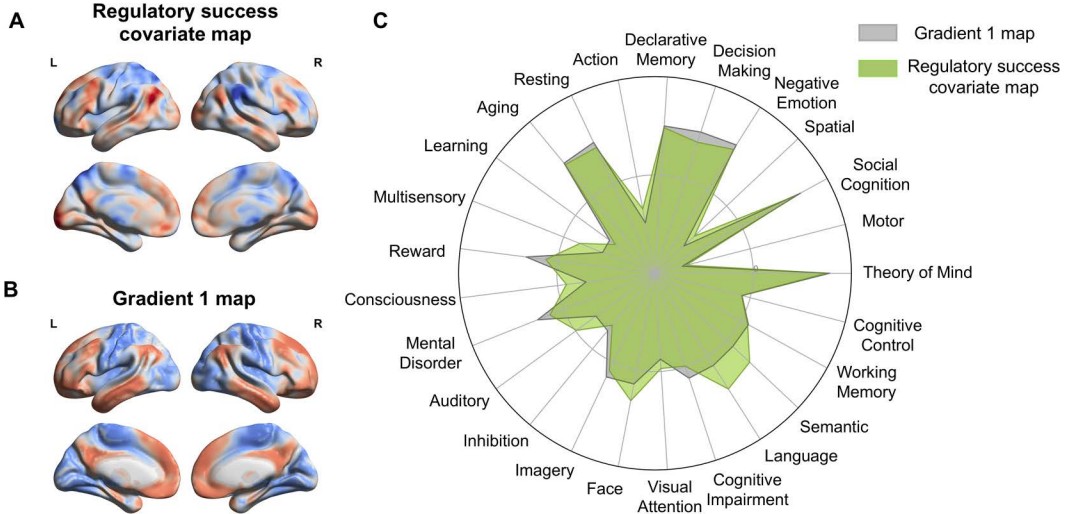

**Fig 4. Meta-analytic functional decoding of associated mental processes (Neurosynth). A.** Covariate map of regulatory success derived from a group-level GLM ($n = 619$) on the emotion regulation task fMRI data (unthresholded $t$-map, red [blue] indicates positive [negative] $t$-values). L/R indicates the left/right hemisphere. **B.** Brain map of the principal Gradient 1 (unimodal and heteromodal cortices), independently derived from Human Connectome Project (HCP) resting-state fMRI data (colors represent loadings of a principal component analysis) [35]. **C.** Visualization of the functional decoding results on 25 Neurosynth topics. Overall, Gradient 1 (gray) and regulatory success covariate maps (green) were characterized by high loadings on declarative memory, decision-making, theory of mind, social cognition, negative emotion, and language-related processes. Brain images were plotted using BrainNet Viewer [58]. The data and code used to generate the figure can be found at: https://osf.io/yk85c/.

neural maps with a large database of meta-analytic maps tagged with cognitive and psychological terms, enabling unbiased inference about the likely mental functions associated with a given brain pattern [60–62]. Because the psychological functions associated with Gradient 1 are already well characterized in the literature [35], our decoding step served as a hypothesis-driven test of whether the established psychological profile of this principal gradient aligned with the profile associated with regulatory success in our joint data. Importantly, using Neurosynth to decode unthresholded whole-brain maps allowed us to characterize the psychological functions reflected in distributed activation patterns rather than isolated clusters, offering an interpretive framework that complements the manuscript's whole-brain perspective.

We applied this method to both the unthresholded group-level covariate map of regulatory success and the independent, canonical Gradient 1 map. Decoded terms for both maps converged on cognitive and emotional domains—including negative emotion, social cognition, memory, imagery, control, language, and attention—highlighting the broad mental processes supporting effective emotion regulation (see radar plots in Fig 4C). High loadings on terms like mental disorder, cognitive impairment, decision-making, or aging also point to the potential clinical relevance and importance for successful aging for both neural maps. This functional overlap of decoded terms provides further support for the notion that Gradient 1—capturing macroscale transitions from unimodal to heteromodal (association) cortices—shares a cognitive signature with the neural substrates underlying individual variability in emotion regulation success. Supplementary decoding results for Gradients 2–5 are shown in S6 Fig, with full term listings in S6 Table.

## Supplemental analyses

**Activation pattern shifts along gradient 1 are associated with negative affect in daily life.** So far, we have demonstrated a robust link between Gradient 1 shifts of global neural activation patterns and regulatory success as captured in a laboratory emotion regulation task. This raises an important question: do gradient-specific neural reconfigurations also reflect self-reported emotional experiences in *daily life*? A subset of RS participants ($n = 55$)

completed a 1-week experience sampling protocol [63], rating their emotions multiple times per day on a 5-point scale (1 = extremely negative, 5 = extremely positive; M ± SD: 3.41 ± 0.35). For these participants, greater regulation-induced shifts of global activation patterns along Gradient 1 observed in the laboratory emotion regulation task were significantly associated with lower negative affect in real life as assessed by experience sampling data ($\beta$ = 1.19, *SE* = 0.38, 95% CI [0.44, 1.94], $t$(47) = 3.13, $p$ = 0.002). No shifts along other gradients were significantly related to self-reported affect in daily life (S7 Table). These findings suggest that stronger neural reconfiguration along Gradient 1 is associated with more positive affect in everyday life, consistent with more effective emotion regulation in those individuals. This finding points towards the ecological validity of our results outside of well-controlled laboratory settings and generalization across task-based and self-reported affective outcomes.

**Gradient-based reconfigurations don't reflect habitual use of emotion regulation strategies.** To assess the specificity of our main result, we also examined the association between gradient-specific shifts in global activation patterns and habitual use of emotion regulation strategies (reappraisal and suppression), as assessed by the ERQ questionnaire completed by both samples [7] (S7 Fig). First, behaviorally, we found no significant correlation between either self-reported habitual use of reappraisal or suppression and task-based regulatory success (all $p$'s > 0.05, corrected). Second, on a neural level, we ran two modified LMMs of Eq. 1 that used questionnaire scores (instead of task-based regulatory success scores) as the dependent variable (joint sample). However, systematic changes of global activation maps along gradients were not associated with either ERQ subscale ($p$'s > 0.17), suggesting specificity of our results for variance in regulatory success using reappraisal, but not its habitual use in daily life (or other emotion regulation strategies like expressive suppression).

**Interactions among gradients.** Supplemental analyses also tested whether accounting for interactions between gradient-specific shifts improved predictions of individuals' regulatory success in the laboratory emotion regulation task. We compared nested linear regression models of increasing complexity—from a main effects model (Eq. 1) to models including all possible two-way, three-way, and higher-order interactions. Model fit, assessed via the Akaike Information Criterion (AIC), consistently favored the main effects model, with no systematic improvement from including interaction terms (S8 Table and S8 Fig).

Second, following prior work [50], we tested whether overall shifts in gradient space—quantified as Euclidean Distance (ED) between "Look" and "Regulate" conditions—could account for variance in the observed regulatory success. Using both five-dimensional and reduced three-dimensional gradient spaces [50], ED-based models showed poorer fit than the main model (S9 Table). Together, these results suggest that regulatory success is best explained by targeted changes in global activation patterns along individual principal gradients (i.e., especially along Gradient 1) rather than broad shifts across multiple gradients.

**Region-wise gradient shifts do not correspond to variance in regulatory success.** As a validation analysis, we assessed whether regional (cortical and subcortical) gradient shifts were related to regulatory success. After FDR correction ($q$ = 0.05), no individual regions showed significant associations with task-based regulatory success scores, suggesting that the whole-brain gradient effect is not driven by isolated regional sources.

## Discussion

A central aim of affective neuroscience is to understand why some individuals regulate their emotions more effectively than others. Using a novel gradient-based framework that captures the full spectrum of neural activity during emotional experience and regulation, we offer new insights into the neural mechanisms underlying variance in emotion regulation success. Unlike traditional approaches that focus on isolated regions or discrete networks, our findings reveal that successful regulation involves systematic reconfigurations of large-scale activation patterns along the brain's intrinsic macroscale organization (gradients). These gradients characterize systematic spatial patterns of large-scale neural activity. In particular, dynamic neural reconfigurations along Gradient 1—which spans from unimodal (sensory) to heteromodal

(association) brain areas—robustly reflected individual differences in regulation success across two large independent samples, as reflected in greater reductions in negative affect during a standardized laboratory reappraisal task. These gradient-based neural reconfigurations were also associated with lower levels of negative affect in daily life, as measured through smartphone-based experience sampling in a subset of participants. This suggests a meaningful link between gradient-based brain dynamics and real-world emotional resilience. By focusing on system-wide neural transitions rather than localized activations or individual networks, this framework enhances our understanding of the neural basis of emotion regulation across contexts. Aligning emotion regulation research with the brain's macroscale architecture lays the foundation for biologically grounded, ecologically valid, and clinically actionable models. In particular, gradient brain dynamics may offer promising neural markers for identifying individuals at risk for affective disorders and targets for personalized interventions such as neurostimulation, neurofeedback, or psychotherapy.

Our findings also have implications for the ongoing debate on the neural separability hypothesis [17], which proposes that emotion generation is primarily subserved by subcortical regions such as the amygdala and insula [64–66], whereas regulation relies on prefrontal and parietal control systems [11,21,67]. Challenging this view, we show that global brain states during emotion generation (look) and regulation (reappraisal) occupy relatively similar positions along principal gradients. This suggests that the two processes are supported—at least in part—by overlapping neural architectures. This spatial alignment likely reflects the integrative nature of emotional experience, where both conditions involve evaluating affective stimuli, directing attention, and constructing subjective feeling states [20,68]. Rather than reflecting activation of distinct neural modules, emotion generation, and regulation may constitute dynamic configurations along a continuum of macroscale functional brain organization. This interpretation aligns with recent univariate and multivariate findings showing substantial overlap in neural substrates for "look" and "reappraisal" conditions [69]. For instance, Bayesian modeling has identified a common frontoparietal-insular network engaged by generating and regulating emotion [54], suggesting that the psychological distinction between these two complex mental processes does not map cleanly onto discrete neural systems.

Although "look" and "regulation" evoked broadly similar global activation patterns, emotion regulation was associated with systematic neural reconfigurations along principal gradients of macroscale functional organization. Gradient 1, in particular, revealed prominent regulation-related shifts, indicating a functional transition from unimodal (sensory) to heteromodal (association) brain areas during reappraisal. Prior decoding studies have established that Gradient 1 indexes a hierarchy of cognitive abstraction, with unimodal cortex supporting perceptual and sensorimotor processing, and heteromodal cortex supporting higher-order, context-rich computations less constrained by immediate sensory input [35]. The observed shift toward heteromodal brain areas during reappraisal aligns with the strategy's demands—constructing alternative appraisals, engaging working memory, manipulating language-based representations, and deploying top-down attention—all of which rely on abstract, integrative processing [16,52]. In contrast, the "look" condition more dominantly engages bottom-up perceptual processing and initial affective appraisal. Thus, Gradient 1 reflects a shift from reactive to reflective processing, a trajectory critical for effective emotion regulation. These results highlight the potential of Gradient 1 as a unifying axis for characterizing neurocognitive transitions between emotion generation and regulation, reframing reappraisal as a system-wide reorientation toward abstraction, integration, and context-sensitive processing.

Critically, shifts along Gradient 1 not only characterized general reappraisal-related brain dynamics but were also associated with individual differences in emotion regulation success. This association was validated through out-of-sample prediction across two large, demographically and geographically diverse cohorts spanning the United States, Canada, Austria, and Germany, enhancing both internal and external validity in line with best practices for reproducible brain–behavior research. Greater gradient-based reconfiguration was also associated with more positive daily affect. This suggests that individuals who exhibit stronger functional shifts along Gradient 1 during reappraisal also regulate emotions more effectively in everyday contexts, yielding more beneficial affective outcomes, reinforcing the real-world relevance

of these gradient-based neural markers. This approach represents a significant advance over traditional univariate or regions-of-interest-based analyses to understand variance in emotion regulation success, which have yielded inconsistent results and often focused narrowly on regions such as the prefrontal cortex or amygdala [23,70,71]. These regionally constrained methods may overlook the distributed and interactive neural mechanisms essential for effective regulation [16,18]. By capturing distributed, large-scale neural dynamics, Gradient 1 offers a more comprehensive lens on the *entire suite* of processes supporting effective emotion regulation. Meta-analytic decoding via Neurosynth supports this interpretation, showing that Gradient 1 tracks a diverse range of functions—such as social cognition, memory, imagery, language, attention, and negative emotion—consistent with the multifaceted nature of emotion regulation. Overall, mapping individual differences in regulatory success onto macroscale neural gradients provides a scalable and generalizable framework for understanding emotion regulation in both laboratory and everyday life.

An important open question is whether neural reconfigurations of whole-brain activation patterns along principal gradients also support regulatory success in other domains. To date, only one prior study has applied a gradient-based framework to another self-regulation context—dietary control [50]. Consistent with our findings, that study showed that individual differences in regulation were predicted by changes in whole-brain activation along principal gradients. However, the direction of the effect differed: successful emotion regulation in our study was associated with larger shifts along a single gradient (Gradient 1), whereas better dietary regulators exhibited smaller shifts across multiple gradients, suggesting more efficient or selective network engagement. This divergence may reflect domain-specific demands—such as regulating emotional responses versus resisting temptations during value-based decisions—or differences in task structure. In the dietary task [50], participants used a range of spontaneous strategies without explicit guidance by experimenters, whereas our study employed a single, instructed approach (reappraisal), pointing to the possibility that gradient dynamics might vary by strategy. Future research should investigate how distinct regulation strategies map onto gradient patterns, and whether individual differences in gradient engagement reflect adaptive, context-sensitive control. Clarifying the conditions under which different neural reconfiguration profiles are advantageous will be essential to understanding how flexible brain network dynamics support successful self-regulation across various domains.

Our findings offer novel insights into the neural basis of emotion regulation; yet, several limitations should be noted, highlighting promising directions for future research. First, our results are limited to cognitive reappraisal. Future work should examine how other regulation strategies—such as distraction, suppression, acceptance, and social sharing [6,72,73]—are represented in gradient space, and whether gradient shifts reflect strategy-specific regulation success. Second, our study focused on the implementation phase of regulation (i.e., applying reappraisal to stimuli), but effective regulation unfolds across multiple phases, including situation selection, attention deployment, and strategy selection [74,75]. Mapping these phases onto gradients may further illuminate how the brain's macroscale architecture supports emotional adaptation over time. Third, it is crucial to extend this framework to clinical populations. Emotion dysregulation is a core feature of many mental disorders, including depression, anxiety, post-traumatic stress disorder, and borderline personality disorder [6,76–78]. Traditional neuroimaging research has emphasized dysfunction in isolated regions (e.g., amygdala, prefrontal cortex, and anterior insula [12,79]), potentially overlooking broader disruptions in hierarchical functional integration. Gradient-based analyses may uncover altered large-scale dynamics in these populations, offering a more holistic view of emotion dysregulation. These gradient-based markers may also prove useful in tracking treatment response to interventions such as regulation training, psychotherapy, neurostimulation, or neurofeedback [80–83]. Moreover, future work integrating gradient approaches with neuromodulation techniques (e.g., TMS/tDCS) or lesion models may offer causal leverage, allowing researchers to test whether directly manipulating specific network states can alter regulatory outcomes. By integrating gradient neuroscience with clinical research, we may advance mechanistic understanding, improve diagnostic precision, and develop targeted, personalized treatments for affective disorders [84].

## Materials and methods

### Ethics statement

All participants gave written informed consent and were paid for their participation. The experimental procedures were approved by the Institutional Review Board at the University of Pittsburgh (study 1 and 2 of the DS), Medical University of Vienna (study 1 of the RS), Freie University of Berlin (study 2–4 of the RS), German Psychological Society (DGP) (study 5 and 6 of the RS), and Health Sciences and Affiliated Teaching Hospitals Research Ethics Board (HSREB) of Queen's University (study 7 of the RS). These experiments were conducted in accordance with the Declaration of Helsinki.

### Participants

This study combines data from two large samples utilizing versions of a well-established emotion regulation task, incorporating both behavioral and fMRI task-based measures: a discovery sample (DS, $n = 358$; 176 females; age $= 42.04 \pm 6.96$) and a replication sample (RS, $n = 263$; 156 females; age $= 25.22 \pm 7.78$). The DS includes participants from two fMRI studies [8,85], previously analyzed together [54]. To validate our DS findings in a large independent cohort, the RS comprises data from seven prior studies on emotion regulation [2,29,3, 55–57,63]. Detailed demographic and study-specific information is presented in Table 3. One participant each from the DS and RS samples was excluded due to outlier regulation success scores that exceeded three standard deviations from the sample mean, as identified using NumPy (v1.23.5) in Python.

### Behavioral paradigm and measures

**Emotion regulation (reappraisal) task.** All participants completed a version of a well-established emotion regulation task during fMRI scanning (Fig 1A) [21,29,57,86]. Each trial began with a visual cue instructing participants to either passively view an image without regulating their emotional response ("Look") or actively decrease negative feelings using cognitive reappraisal ("Regulate"). The discovery sample also included a condition in which participants passively viewed neutral images ("Look Neutral") [54], which was excluded from our study due to its absence in other datasets and our focus on variation in emotion regulation success. Following the cue, a negative image was presented for up to 8 s (see Table 3, Regulation Duration, for sub-sample details) and participants implemented the instruction ("Look" or "Regulate"). Images were selected from standardized emotional image databases, including the International Affective Picture System

**Table 3. Demographic and task information for each study in the discovery and replication samples.**

| Study | Reference | N (female) | Mean age (SD) | Study location | Regulation duration | Total trial number |
|---|---|---|---|---|---|---|
| *Discovery sample (DS)* | [54] | 358 (176) | 42.04 (6.96) | | | |
| 1 | [85] | 182 (87) | 43.45 (7.24) | USA | 7 s | 45 |
| 2 | [8] | 176 (89) | 40.58 (6.31) | USA | 7 s | 45 |
| *Replication sample (RS)* | | 263 (156) | 25.22 (7.78) | | | |
| 1 | [55] | 25 (21) | 22.8 (3.30) | Austria | 8 s | 240 |
| 2 | [56] | 37 (31) | 22 (2.58) | Germany | 8 s | 144 |
| 3 | [2] | 35 (29) | 23.17 (3.44) | Germany | 4 s | 140 |
| 4 | [3] | 29 (13) | 24.52 (4.25) | Germany | 4 s | 96 |
| 5 | [29] | 23 (12) | 25.70 (5.95) | Germany | 8 s | 168 |
| 6 | [57] | 59 (20) | 32.47 (11.25) | Germany | 8 s | 90 |
| 7 | [63] | 55 (30) | 21.38 (5.10) | Canada | 6 s | 80 |

*Note.* SD = standard deviation.

[87] (all studies, except study 7 in the RS dataset) or the Nencki Affective Picture System [88] and Emotional Picture Set [89] (study 7 in the RS, Table 3). After a variable inter-stimulus interval (ISI; 1–8 s), participants rated their current emotional state on a rating scale (see S1 Table for study-specific details on rating scales). Average ratings from all trials in the "Look" condition served as an index of participants' emotional reactivity (available on OSF, https://osf.io/yk85c/; summary descriptive statistics are presented in S1 Table), which was used as a control variable in the neural prediction of emotion regulation success (see below). Emotional state ratings in both task conditions also enabled us to quantify variance in people's emotion regulation success behaviorally, as explained next.

**Emotion regulation success.** Individual differences in emotion regulation success were assessed using a regulatory success score, calculated as the difference in each participant's mean emotional state ratings between "Regulate" and "Look" conditions in the emotion regulation task (Fig 1A). Higher positive scores reflected greater effectiveness in reducing negative affect through reappraisal. To account for variations in rating scale ranges across individual studies, raw scores were normalized (z-scored) within each study to ensure comparability when integrated into samples (DS, RS). The regulatory success score was used as the primary dependent variable in our regression analyses of interest (see below).

**Emotion regulation questionnaire (ERQ).** Participants also completed the ERQ [7], which measures the habitual use of reappraisal and suppression. While ERQ subscales were not part of the primary analyses, they served as a supplemental check to examine associations with task-based regulatory success. Additionally, they allowed us to test whether neural shifts along principal brain gradients (e.g., Gradient 1) were also linked to individual differences in habitual use of emotion regulation strategies.

**Emotional experiences in daily life (experience sampling).** To assess emotional states in daily life, a subset of participants from the RS dataset (Study 7, $n = 55$) completed a week-long experience sampling protocol via the Samply smartphone app [90]. Participants received five randomly timed prompts per day between 9:00 AM and 9:00 PM (minimum 2 hours apart) to complete a brief survey [91–94]. A total of 1,717 surveys were completed (Mean ± SD: $31.31 ± 3.94$ per participant, max = 35). In each survey, participants rated their emotional state prior to the phone-based prompt on a 5-point Likert scale (1 = extremely negative, 5 = extremely positive). Participant-level average ratings (M ± SD: $3.41 ± 0.35$) were used as the outcome variable to test whether task-evoked neural shifts along principal brain gradients in the laboratory emotion regulation task were also associated with individual differences in emotional experiences in daily life.

## MRI data analysis

For both samples (DS and RS), we used GLM-based contrast maps—publicly available for DS (https://neurovault.org/collections/16266/) and shared by the original authors for RS (study 7 available on OSF, https://osf.io/yk85c/; studies 1–6 of the RS are available upon request). Below, we summarize the fMRI acquisition, preprocessing, and GLM procedures; additional details can be found in the original publications (Table 3) and S10 Table.

**Acquisition and preprocessing.** S10 Table summarizes key acquisition parameters for functional MRI data across both samples and preprocessing steps. All brain data were originally preprocessed using statistical parametric mapping software in MATLAB (SPM12 and SPM8; http://www.fil.ion.ucl.ac.uk/spm). Despite some minor variations, preprocessing generally involved slice time correction to account for acquisition time variation; spatial realignment to the mean image with a six-parameter rigid body transformation; co-registration to the individual T1-weighted anatomical image; co-registered images were then normalized to standard Montreal Neurological Institute (MNI) space; and spatial smoothing.

**GLM.** To characterize whole-brain activation patterns in different conditions of the emotion regulation task ("Look" and "Regulate"), for each participant, a GLM was estimated that included the following regressors: 1) R1: picture-viewing period for all "Look" trials, 2) R2: picture-viewing period for all "Regulate" trials, 3) R3: instruction cue period, and 4) R4: emotional state rating period. The main regressors were modeled separately with a boxcar function and convolved with the canonical hemodynamic response function (HRF) in SPM. The duration for R1–R4 corresponded to the study-specific

duration for which each phase was presented to participants in the emotion regulation task (see Table 3 for study-specific durations of regressors of interest, R1 and R2). In addition, the nuisance regressor includes the six motion parameters estimated during the preprocessing of functional brain data (three for translations and three for rotations) and a run-wise constant (if applicable). For an overview of further details on study-specific idiosyncrasies in GLMs across both samples (DS, RS), see the overview at OSF (https://osf.io/yk85c/). For each participant, this GLM allowed us to create two separate contrast images of "Look" or "Regulate" against an implicit baseline as implemented in SPM. These two contrast maps (unthresholded) represent the participant-specific whole-brain activation patterns during natural and regulated emotional experiences, which were projected into the gradient space (see below).

We also set up an additional GLM at the group-level to derive a covariate map of variance in regulatory success across all participants of the joint sample ($n = 619$). Here, we used participant-specific contrast maps of estimated brain responses for "Regulate" (R2)> "Look" (R1) (from the participant-level GLMs described above). This group-level GLM also included individuals' estimated behavioral regulatory success scores as a covariate in the model, which allowed us to obtain an unthresholded covariate contrast map for variance in regulatory success across all participants (Fig 4A).

Another supplemental GLM at the group-level matched the one above, but used participants' Gradient 1 shifts (ΔG1, see below for details) as a covariate in the model. This group-level GLM enabled us to derive a covariate map of variance in shifts along gradient 1 (S3 Fig).

### Gradient analysis

**Principal gradients of macroscale brain organization.** Gradients capture core topographic principles of macroscale brain organization, where regions with similar functional profiles are positioned closer along principal axes of variance [41,95]. To examine how emotion regulation success varies within this framework, we utilized previously established gradient maps derived from 1-hour resting-state fMRI data from the HCP [35,51]. These maps are independent of our participant cohorts (DS, RS) and emotion regulation task and have previously been used as an intrinsic coordinate system of topographical brain organization in which regulatory processes unfold [50]. We focused on the first five gradients, which together explain ~55% of the variance in the independent HCP data. Gradient maps encompassed both cortical and subcortical regions, reflecting the involvement of distributed neural networks of cortical and subcortical areas in emotion processing [11,18,52,64]. Gradient 1 captures the principal axis of variation that distinguishes unimodal sensory regions from heteromodal (association) cortices. Gradient 2 differentiates visual from sensorimotor networks. Gradient 3 dissociates the DMN from the frontoparietal (FP) control network. Gradient 4 separates ventral and dorsal attention networks. Gradient 5 differentiates the visual cortex and ventral attention network. See Fig 2B for an illustration of these principal gradient maps.

**Projecting task-evoked global activation patterns into gradient space.** Following prior work [50], we projected task-evoked whole-brain activation patterns from the "Look" and "Regulate" phases of the emotion regulation task into gradient space (Fig 2A–C). First, for each participant and condition, we computed the Pearson correlation between their GLM-derived, unthresholded contrast image (Look > Baseline or Regulate > Baseline) and each of the five principal gradient maps. Notably, all computations were based on whole-brain principal gradient maps that included both cortical and subcortical voxels. These similarity scores were Fisher *z*-transformed (using the *arctanh* function from *NumPy* in Python) and served as each participant's position along each gradient, separately for each condition ("Look" and "Regulate"). Second, to assess an individual's regulation-related change, we computed difference scores between the "Regulate" and "Look" projections for each gradient (yielding five gradient-specific change scores per participant $i$, $\Delta G1_i$ to $\Delta G5_i$). These individual-level change scores in gradient similarity were then used as predictor variables in a linear mixed model (see below) to examine their associations with individuals' regulatory success observed in the emotion regulation task (see Eq. 1 below).

For supplemental analyses, we also calculated the ED between projected "Look" and "Regulate" activation patterns in the five-dimensional gradient space and a reduced three-dimensional space (G1–G3) [50] (using the *euclidean* function from *Scipy* in Python). This metric captures overall shifts in multidimensional neural configuration instead of changes along individual gradients. These distance metrics were used in supplemental analyses to assess whether broader reconfiguration of regulation-related activation patterns across multiple gradients is associated with regulatory success, beyond changes along a single gradient.

**Explaining variance in regulatory success from shifts in gradient-wise similarity.** To test if individual differences in emotion regulation success are associated with systematic reconfigurations of global activation maps along principal gradients of the brain's topographical organization, we estimated a LMM using the *lmer* function in R (version 4.3.3) [96]. Participant-specific regulatory success scores observed in the emotion regulation task served as the dependent variable (Eq. 1). Predictors included participants' "Regulation"-related changes in whole-brain activation patterns compared to "Look" along each principal gradient (ΔG1 to ΔG5). We also controlled for participants' age, sex at birth, and emotional reactivity (average emotional rating during "Look" trials) (fixed effects) and included study ID as a random effect.

$$\text{Regulatory Success} = \beta_0 + \beta_1 \Delta G_1 + \beta_2 \Delta G_2 + \beta_3 \Delta G_3 + \beta_4 \Delta G_4 + \beta_5 \Delta G_5$$
$$+ \beta_6 \text{Emotional Reactivity} + \beta_7 \text{Age} + \beta_8 \text{Sex} + \beta_9 \text{Study ID} + \varepsilon \qquad (1)$$

**Out-of-sample predictions.** To evaluate the robustness and generalizability of our main model (Eq. 1), we performed two out-of-sample prediction analyses. First, we implemented leave-one-participant-out cross-validation within each sample (DS, RS, and, for completeness, the joint sample). In each iteration, one participant was excluded; the model was then trained on the remaining participants and used to predict the excluded participant's regulatory success score, as implemented using "*lmer*" and "*predict*" functions in R (v4.3.3). Model performance was assessed by correlating predicted and observed regulatory scores across all left-out participants. Statistical significance was determined using a permutation test (5,000 iterations), comparing the observed prediction accuracy to an empirical null distribution of accuracies achieved by chance, generated by shuffling regulatory success scores. *P*-values were Bonferroni-corrected for multiple comparisons [59]. Second, to test cross-sample generalizability, we trained the model on DS data and predicted the regulatory success in all RS participants. This procedure followed the same evaluation and permutation testing approach as above. Together, these analyses assessed the predictive power of task-evoked shifts in activation patterns along principal gradients both within and across independent cohorts.

## Neurosynth functional decoding

Next, we examined and compared the underlying mental process associated with principal Gradient 1 (Fig 4B) and the activation pattern that covaries with variance in regulatory success (Fig 4A). To this end, we implemented a meta-analytic functional decoding analysis using the Neurosynth database [53], following previous applications of this method for functional interpretation of univariate or gradient-based brain maps [35,97,98]. Specifically, we used the NiMARE toolbox (Neuroimaging Meta-Analysis Research Environment) in Python (version 3.8) to compare our unthresholded group-level *t*-map (covariation with regulatory success across participants) to thousands of reverse inference maps in the Neurosynth database [99]. We also repeated the approach using the canonical Gradient 1 map. The analyses used the default Neurosynth term set (available at https://github.com/neurosynth/neurosynth-data) and reverse inference maps, after excluding noncognitive terms and brain structures such as 'stimulation', 'method', 'cortex', or 'gyrus' (yielding 25 terms; see S6 Table for a complete list). This decoding procedure calculates Pearson correlation coefficients between an input brain map and each term-based meta-analytic map, yielding a ranked list of cognitive terms most strongly associated with the observed neural pattern [54,60,61]. This approach enables meta-analytical, data-driven inference about the likely mental processes underlying spatial activation patterns, based on established psychological constructs.

## Supplemental models

First, to assess the ecological validity of our findings, we examined whether gradient-based changes in regulation-related neural activation also reflect affective outcomes in daily life. Specifically, we applied a version of Eq. 1 using average self-reported affect—collected via smartphone-based experience sampling over 1 week—as the dependent variable in a subset of participants ($n = 55$, RS Study 7).

Second, to test the specificity of our results, we examined whether gradient-based changes in regulation-related neural activation patterns also reflect habitual use of reappraisal (or suppression), as assessed in the established ERQ questionnaire. We ran two versions of the main model (Eq. 1) using ERQ subscales for habitual cognitive reappraisal and expressive suppression as the dependent variable.

Third, to confirm that predictive effects stemmed from shifts along individual gradients rather than complex multi-gradient trajectories, we conducted several post hoc analyses. Here, we first tested modified versions of Eq. 1 that included interaction terms among gradient-specific change scores ($\Delta G1$ to $\Delta G5$). Using a series of nested models, we progressively added two-way, three-way, and higher-order interactions, up to a full model including a five-way interaction. Model fit was evaluated using the AIC [100], as implemented via the *performance* package in R (v4.3.3) [101]. Lower AIC values indicated better model fit.

Fourth, we replaced the five gradient-specific change scores with a single metric—$\Delta ED$—capturing the overall shift between "Look" and "Regulate" conditions in five-dimensional gradient space, estimated using the *euclidean* function from *Scipy* in Python) (Eq. 2). We also tested an analogous model using a reduced 3-D gradient space based on the first three principal gradients (~50% explained variance) [50]. Model fit was again evaluated using the AIC [100], as implemented via the *performance* package in R (v4.3.3), with lower AIC values indicating better model fit. [101].

$$\text{Regulatory Success} = \beta_0 + \beta_1 \Delta ED + \beta_2 \text{Emotional Reactivity} + \beta_3 \text{Age} + \beta_4 \text{Sex} + \beta_5 \text{Study ID} + \varepsilon \qquad (2)$$

Fifth, to further assess whether large-scale gradient shifts, rather than effects confined to specific regions, account for individual differences in regulatory success, we conducted a supplemental *region-wise* analysis. We extracted region-specific gradient shifts from both cortical and subcortical areas and examined their associations with individual differences in task-based regulatory success. For the cortical analysis, we used the Schaefer-400 parcellation, consistent with prior region-specific gradient work [102,103], and we used the Harvard–Oxford Subcortical Atlas [104] to define 14 subcortical regions. We then estimated a LMM in which region-specific gradient shifts were entered as predictors of individual regulatory success, adjusting for age, sex assigned at birth, and emotional reactivity, and including study ID as a random effect. Because this model was repeated across all regions, we applied a false discovery rate (FDR) correction to account for multiple comparisons, performing the correction separately for cortical (400 regions) and subcortical (14 regions) regions.

## Supporting information

**S1 Fig. Histograms of study-specific, unstandardized regulatory success scores in the discovery and replication samples.** Higher positive values indicate more effective regulation of task-evoked emotional states (ratings in "Regulate" – "Look"). See S1 Table for an overview of the study-specific rating scales. Data underlying this figure can be found at: https://osf.io/yk85c/. (TIF)

**S2 Fig. Distributions of similarity scores for "Look" (grey) and "Regulate" (purple) activation patterns across gradients.** Dotted lines show condition-specific mean similarities with a principal gradient; asterisks indicate significant differences between both condition-specific gradient similarity scores along a particular gradient (Wilcoxon signed-rank tests, *$p < 0.05$; **$p < 0.01$; ***$p < 0.001$; Bonferroni-corrected). Data underlying this figure can be found at: https://osf.io/yk85c/. (TIF)

**S3 Fig. Covariate map of Gradient 1 shift.** *Left:* Principal Gradient 1 map derived from independent resting-state fMRI data in the Human Connectome Project (HCP), depicting the unimodal–to–heteromodal cortical hierarchy for comparison. *Right:* Covariate *t*-map showing brain regions in which task-evoked activation during emotion regulation (Regulate > Look) covaries with participant-specific Gradient 1 shifts (ΔG1), thresholded at $p < 0.001$ (uncorrected) at the voxel level and cluster level FWE-corrected at $p < 0.05$. Warm (red) colors reflect positive associations; cool (blue) colors reflect negative associations. The results indicate that larger Gradient 1 shifts during emotion regulation are associated with greater activation in heteromodal regions and reduced activation in unimodal regions. Brain maps underlying this figure can be found at: https://osf.io/yk85c/.
(TIF)

**S4 Fig. Study-specific prediction of regulatory success (leave-one-participant-out cross-validation) based on gradient-wise changes of global activation maps (Eq. 1).** Data underlying this figure can be found at: https://osf.io/yk85c/.
(TIF)

**S5 Fig. Cross-sample prediction of regulatory success based on systematic reconfigurations of global activation maps along principal gradients of topographical organization. A.** Pearson correlations between observed and predicted regulatory success scores using cross-sample prediction ($r = 0.37$, 95% CI [0.26, 0.47], $p_{perm} < 0.001$), where we trained the predictive model on discovery sample data (DS) and applied the model on replication sample data (RS) to assess the out-of-sample generalizability. The statistical significance was assessed using a nonparametric permutation test. **B.** Visualization of the observed predictive accuracy of the out-of-sample prediction (dotted line) compared to the empirical null distribution of the prediction observed by chance (histogram). Data underlying this figure can be found at: https://osf.io/yk85c/.
(TIF)

**S6 Fig. Neurosynth meta-analytic functional decoding of Gradients 2–5 and overlap with results for the covariate map of individual differences in emotion regulation success.** Data underlying this figure can be found in S6 Table.
(TIF)

**S7 Fig. Histograms of ERQ (emotion regulation questionnaire) scores for the habitual use of reappraisal and suppression.** Mean ± SD [range]; ERQ-cognitive reappraisal: discovery sample: 29.86 ± 5.59; replication sample: 19.63 ± 7.25; joint sample: 25.56 ± 8.06; ERQ-expressive suppression: discovery sample: 14.87 ± 4.96; replication sample: 18.36 ± 4.90; joint sample: 16.35 ± 5.23. Data underlying this figure can be found at: https://osf.io/yk85c/.
(TIF)

**S8 Fig. AIC (Akaike Information Criterion) values for model comparisons.** Across all samples, the main model (Eq. 1) showed the lowest AIC, indicating that it provided the best fit relative to models including interaction terms. Data underlying this figure can be found at: https://osf.io/yk85c/.
(TIF)

**S1 Table. Study-specific emotional state ratings and regulatory success.**
(DOCX)

**S2 Table. Dataset-specific statistical tests of projected activation maps against zero per principal gradient.**
(DOCX)

**S3 Table. Dataset-specific statistical tests of projected whole-brain activation maps in "Look" versus "Regulate" per principal gradient.**
(DOCX)

**S4 Table. Significant clusters showing covariation between Gradient 1 Shifts (ΔG1) and task-evoked activation (Regulate > Look).**
(DOCX)

**S5 Table. Significant clusters showing covariation between participants' regulatory success and task-evoked activation (Regulate > Look).**
(DOCX)

**S6 Table. Neurosynth meta-analytical decoding results.**
(DOCX)

**S7 Table. Regulation-related shifts of activation patterns along Gradient 1 in the laboratory emotion regulation task predict average negative affect in daily life.**
(DOCX)

**S8 Table. AIC values of model comparisons (models with interaction terms).**
(DOCX)

**S9 Table. AIC values of the model comparisons (models with ED, Euclidean distance).**
(DOCX)

**S10 Table. Overview of the study-specific MRI acquisition parameters and preprocessing of fMRI data.**
(DOCX)

## Author contributions

**Conceptualization:** Ruien Wang, Anita Tusche.

**Data curation:** Ruien Wang, Carmen Morawetz.

**Formal analysis:** Ruien Wang, Remi Janet.

**Funding acquisition:** Anita Tusche.

**Investigation:** Ruien Wang, Carmen Morawetz, Anita Tusche.

**Methodology:** Ruien Wang, Remi Janet.

**Project administration:** Ruien Wang, Anita Tusche.

**Resources:** Carmen Morawetz.

**Software:** Ruien Wang, Remi Janet.

**Supervision:** Anita Tusche.

**Visualization:** Ruien Wang, Remi Janet, Carmen Morawetz.

**Writing – original draft:** Ruien Wang, Carmen Morawetz, Anita Tusche.

**Writing – review & editing:** Remi Janet, Carmen Morawetz, Anita Tusche.

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
