## [Editor Report · Decision Letter 0]

11 Aug 2025

Dear Dr Wang,

Thank you for submitting your manuscript entitled "Emotion Regulation in the Gradient Framework: Large-Scale Brain Organization Shapes Individual Differences in Reappraisal Success" for consideration as a Research Article by PLOS Biology.

Your manuscript has now been evaluated by the PLOS Biology editorial staff as well as by an academic editor with relevant expertise and I am writing to let you know that we would like to send your submission out for external peer review.

Once your full submission is complete, your paper will undergo a series of checks in preparation for peer review. After your manuscript has passed the checks it will be sent out for review. To provide the metadata for your submission, please Login to Editorial Manager (https://www.editorialmanager.com/pbiology) within two working days, i.e. by Aug 13 2025 11:59PM.

Kind regards,

Christian

Christian Schnell, PhD

Senior Editor

PLOS Biology

cschnell@plos.org

---

## [Decision Letter · Decision Letter 1]

3 Oct 2025

Dear Dr Wang,

Thank you for your patience while your manuscript "Emotion Regulation in the Gradient Framework: Large-Scale Brain Organization Shapes Individual Differences in Reappraisal Success" was peer-reviewed at PLOS Biology. It has now been evaluated by the PLOS Biology editors, an Academic Editor with relevant expertise, and by several independent reviewers.

In light of the reviews, which you will find at the end of this email, we would like to invite you to revise the work to thoroughly address the reviewers' reports.

As you will see below, the reviewers think that the study is overall well executed and provides important insights, but they also raise concerns about the clarity of the presentation and the (potentially) exploratory type of the analysis and that the claims regarding the involvement of the entire brain are currently not sufficiently supported. We therefore ask you to address the reviewer concerns by improving clarity of the presentation, transparency of the analysis, and add the analyses as suggested by Reviewer 3.

Given the extent of revision needed, we cannot make a decision about publication until we have seen the revised manuscript and your response to the reviewers' comments. Your revised manuscript is likely to be sent for further evaluation by all or a subset of the reviewers.

**IMPORTANT - SUBMITTING YOUR REVISION**

*Re-submission Checklist*

*Published Peer Review*

*PLOS Data Policy*

*Blot and Gel Data Policy*

Sincerely,

Christian

Christian Schnell, PhD

Senior Editor

PLOS Biology

cschnell@plos.org

REVIEWS:

Reviewer #1: In this work, the authors applied spatial gradient analysis to brain activation during emotion regulation and found that shifts along a gradient differentiating unimodal and heteromodal brain areas can predict reappraisal success. This provides a novel angle for understanding emotion regulation success within a network framework. The study is well-powered, and the out-of-sample prediction performance is impressive. The writing is clear, and the analyses are clean. I have a few suggestions.

1. The interpretation that large-scale brain organization "shapes" individual differences in reappraisal success seems to imply causality, whereas the analysis is correlational.

2. The authors provide a good explanation of the processes underlying gradient shifts, largely drawing from HCP data where these gradients were originally defined. However, in the context of emotion regulation, readers may still want to know which specific brain regions' activation (Regulate vs. Look) is related to gradient-1 shifts. For example, is VLPFC activation or amygdala deactivation correlated with gradient shifts?

3. Emotion reactivity is included as a covariate predicting reappraisal success and emerges as a significant predictor. Could the authors elaborate on how they interpret this? Additionally, is emotion reactivity correlated with gradient-1? If so, this could influence the interpretation of the gradient-1 findings.

4. In the discussion the authors state: "our findings also have implications for the neural separability hypothesis [17]." This phrasing suggests their work represents the hypothesis itself, rather than engaging with the ongoing debate. A more precise formulation would be: "our findings also have implications for the ongoing debate on the neural separability hypothesis [17]."

5. Figure 2D is difficult to visualize. The distributions of 'Look' and 'Regulate' appear very similar, and the dashed line does not make it clear which condition shows higher similarity with the gradients. Since most emotion regulation studies focus on the contrast between 'Look' and 'Regulate,' why not show the similarity of this contrast with the gradients, or alternatively present the distribution of ΔG?

6 Figure S4 only shows positive neural correlates of reappraisal success. Were any regions significant as negative correlates?

Reviewer #2: This paper examined brain activation patterns that may indicate emotion regulation. The authors examined two large fMRI datasets that employed an emotion regulation task, and they examined principal gradients based on resting state data from the Human Connectome Project. The authors identified a gradient that was associated with emotion regulation success. The association was present in both datasets, and also was associated with negative affect in daily life in a subset of participants. The approach was novel and the question is of interest to researchers in the field. There were several issues that reduced my enthusiasm for the paper.

1. The gradient approach was interesting and novel but it was unclear the extent that it was exploratory versus hypothesis driven. It was also unclear why this approach was chosen over other approaches that similarly consider whole brain functional hierarchies. Further, given the reverse inference approach of the neurosynth analysis, the overall impression of this was an exploratory approach that could have worked out any number of ways. Exploratory analyses have value but the presentation did not clearly communicate the extent that the approaches were pre-determined or post hoc. Were you truly agnostic about which of the gradients to expect as significant?

2. The neurosynth exploration for theory-neutral explanations seems at odds with the premise of the neural systems topographical landscape. The coordinates reflect task based activations that survive multiple threshold corrections, etc, but do not reflect a whole brain assessment of how the brain accomplishes the tasks associated with the coordinates. This approach seems to contradict the premise of the rest of the paper—that the whole brain could contribute in hierarchical ways.

3. The conclusions seem to overstate the results. "These findings advance a network-level account of regulatory success, offering a biologically grounded, ecologically valid framework for understanding adaptive emotional functioning" This seems to imply that previous imaging approaches were not biologically grounded? Or were not ecologically valid? I did not find compelling evidence presented to reach those same conclusions. Also, the "network level account" seems to overstate the approach—the network function was not probed independently of the task to determine the extent that it contributed to emotion function. It seemed context dependent. Activation patterns within a network framework seems necessary to establish the network level conclusion, but not sufficient. It seems that further probing would be required to rule out alternative explanations.

Reviewer #3 (Boris Bernhardt signed his report): Emotion Regulation in the Gradient Framework: Large-Scale Brain Organization Shapes Individual Differences in Reappraisal Success

The authors projected global fMRI activation patterns from an emotion regulation (ER) lab task onto the principal sensory-transmodal gradient (G1) from the HCP datasets. In both datasets, individual differences in ER success were predicted by systematic differences along G1. Gradient based reconfiguration also predicted lower negative affect in daily life as measured with a smartphone app. Neurosynth meta-analysis revealed that G1 and ER success align with social cognition, memory, attention, and negative emotions. The results are promising indicators that gradient based coordinate systems may help to better understand the network basis of emotional and regulatory processes in the brain, and could be informative for the design of brain based biomarkers.

This is overall a well written and well structured paper. Methods appear state-of-the-art and the study features an independent replication as well as generalization to smartphone real life behavioral prediction. I have the following somewhat minor comments for the authors to improve on their already accomplished work:

1) In the introduction, the authors may want to briefly discuss some conceptual and empirical work from Lisa Feldman Barrett's group that connected gradient research with emotional and allostatic processes (https://pmc.ncbi.nlm.nih.gov/articles/PMC11117115/). Other work that related gradients to information processing hierarchies in the human brain, in both healthy and diseased populations, may be relevant as it can help to dissociate more sensory from higher order integrative and regulatory processes.

2) Page 8 - the authors suddenly mention 5 gradients. It might be worthwhile to also describe these 5 gradients a bit more in the introduction and also the results, as not everyone may have an intuitive understanding of how they partition variance across the cortica; surface (eg in terms of sensory-transmodal, visual-somatomotor, multiple demand, etc)…They kinda do this on P.9, so there could be earlier mention of them I find.

3) Can the authors clarify in the methods/findings whether findings were derived from cortical data only, or whether cortical and subcortical data were analyzed together. If the former, a brief justification may help and/or a demonstration of consistency when all GM voxels are analyzed/mapped.

4) Its mainly semantics, but I would refrain from using the term 'predict' when reporting the LMM findings. For within sample correlations, 'Explain' or 'Correlate' is possibly more accurate but they can further emphasize the DS to RS prediction in the findings and abstract.

5) Is neurovault still a thing? They could deposit the unthresholded maps there as well in addition to osf.

6) This final point is more exploratory, but after reading the discussion, I wondered whether similar findings could emerge if the analyses were confined to a specific region previously implicated in emotion regulation and emotional reactivity—such as the amygdala—rather than using whole-cortex data. For example, one could derive region-specific gradients and test whether shifts in activity within that space correlate with ER success and predict individual differences. If such effects do not emerge at the regional level, it could further support the idea that large-scale, cortex-wide organization is critical for understanding emotion regulation.

---

## [Decision Letter · Decision Letter 2]

23 Jan 2026

Dear Dr Wang,

Thank you for your patience while we considered your revised manuscript "Emotion Regulation in the Gradient Framework: Large-Scale Brain Organization Reflects Individual Differences in Reappraisal Success" for publication as a Research Article at PLOS Biology. This revised version of your manuscript has been evaluated by the PLOS Biology editors, the Academic Editor and two the original reviewers.

Based on the reviews, we are likely to accept this manuscript for publication, provided you satisfactorily address the following data and other policy-related requests:

* We would like to suggest a different title to improve its accessibility for our broad audience:

Emotion regulation involves systematic gradient-based reconfigurations of large-scale activation patterns in the human brain

* Please add the links to the funding agencies in the Financial Disclosure statement in the manuscript details.

* Please include the approval/license numbers of the ethical approval for the experiments.

* DATA POLICY:

Regardless of the method selected, please ensure that you provide the individual numerical values that underlie the summary data displayed in the figure panels as they are essential for readers to assess your analysis and to reproduce it.

* CODE POLICY

Per journal policy, if you have generated any custom code during the course of this investigation, please make it available without restrictions. Please ensure that the code is sufficiently well documented and reusable, and that your Data Statement in the Editorial Manager submission system accurately describes where your code can be found. More information on our Code Policy, what and how to share can be found here: https://journals.plos.org/plosbiology/s/code-availability

We expect to receive your revised manuscript within two weeks.

*Published Peer Review History*

*Press*

Sincerely,

Christian

Christian Schnell, PhD

Senior Editor

cschnell@plos.org

PLOS Biology

Reviewer remarks:

Reviewer #2: Thank you for the thoughtful attention to all the issues raised previously. This version reads better and seems like a valuable contribution.

Reviewer #3 (signed as Boris Bernhardt): The authors have been very responsive to all the reviews and the paper is now ready for publication.

---

## [Editor Report · Decision Letter 3]

6 Feb 2026

Dear Ruien,

Thank you for the submission of your revised Research Article "Emotion Regulation Success Involves Systematic Gradient-based Reconfigurations of Large-scale Activation Patterns in the Human Brain" for publication in PLOS Biology. On behalf of my colleagues and the Academic Editor, Yina Ma, I am pleased to say that we can in principle accept your manuscript for publication, provided you address any remaining formatting and reporting issues. These will be detailed in an email you should receive within 2-3 business days from our colleagues in the journal operations team; no action is required from you until then. Please note that we will not be able to formally accept your manuscript and schedule it for publication until you have completed any requested changes.

PRESS

Sincerely,

Christian

Christian Schnell, PhD

Senior Editor

PLOS Biology

cschnell@plos.org